# 8-Oxoguanine Disrupts G-Quadruplex DNA Stability and Modulates FANCJ AKKQ Peptide Binding

**DOI:** 10.3390/molecules30163424

**Published:** 2025-08-20

**Authors:** Laura Campbell, Kaitlin Lowran, Emma Cismas, Colin G. Wu

**Affiliations:** 1Department of Biochemistry, University of Wisconsin-Madison, Madison, WI 53706, USA; lcampbell22@wisc.edu; 2Department of Chemistry, Oakland University, Rochester, MI 48309, USA; hx9909@wayne.edu (K.L.); ecismas@oakland.edu (E.C.); 3Institute of Environmental Health Sciences, Wayne State University, Detroit, MI 48202, USA

**Keywords:** G-quadruplex, 8-oxoguanine, circular dichroism, FANCJ, helicase, DNA repair, fluorescence spectroscopy

## Abstract

Guanine-rich nucleic acid sequences can adopt G-quadruplex (G4) structures, which pose barriers to DNA replication and repair. The FANCJ helicase contributes to genome stability by resolving these structures, a function linked to its G4-binding site that features an AKKQ amino acid motif. This site is thought to recognize oxidatively damaged G4, specifically those containing 8-oxoguanine (8oxoG) modifications. We hypothesize that FANCJ AKKQ recognition of 8oxoG-modified G4s (8oxoG4s) depends on the sequence context, the position of the lesion within the G4, and overall structural stability. Using fluorescence spectroscopy, we measured the binding affinities of a FANCJ AKKQ peptide for G4s formed by (GGGT)_4_, (GGGTT)_4_, and (TTAGGG)_4_ sequences. G4 conformation and thermal stability were assessed by circular dichroism spectroscopy. Each sequence was modified to include a single 8oxoG at the first (8oxo1), third (8oxo3), or fifth (8oxo5) guanine position. In potassium chloride (KCl), the most destabilized structures were (GGGT)_4_ 8oxo1, (GGGTT)_4_ 8oxo1, and (TTAGGG)_4_ 8oxo5. In sodium chloride (NaCl), the most destabilized were (GGGT)_4_ 8oxo1, (GGGTT)_4_ 8oxo5, and (TTAGGG)_4_ 8oxo5. FANCJ AKKQ binding affinities varied according to damage position and sequence context, with notable differences for (GGGT)_4_ in KCl and (TTAGGG)_4_ in NaCl. These findings support a model in which FANCJ binding to G4 and 8oxoG4 structures is modulated by both the oxidative damage position and the G4 local sequence environment.

## 1. Introduction

G-quadruplexes (G4s) are stable secondary structures that are formed by guanine-rich nucleic-acids. G4 DNA structures arise from intramolecular Hoogsteen hydrogen bonding within single-stranded (ss) DNA containing four tracts of three or more consecutive guanines [1]. Monovalent cations further stabilize G4s by centrally coordinating the guanine tetrads, with potassium (K^+^) and sodium (Na^+^) ions being the most biologically relevant. Among them, potassium ions generally confer greater stability to G4s [2,3,4]. 

The canonical sequence motif for G4 formation is (GGGN_1-7_)_4_, where N denotes any intervening nucleotides [5]. This has led to an estimate of over 400,000 potential G4 forming sequences being identified in the human genome [6]. G4s can adopt multiple topologies—parallel, antiparallel, or hybrid—depending on the nucleotide sequence and solution conditions (Figure 1) [4,7]. These G4 conformations are defined by how the loop regions (N1-7) connect the guanine tracts [8,9] and can be distinguished using circular dichroism (CD) spectroscopy [10]. The spatial arrangements of G4s are biologically significant, as many G4-binding proteins have a preference for specific topologies; the Pif1 helicase, for example, binds and unfolds antiparallel G4s [11,12,13,14,15]. 

G4 structures are enriched in genomic regions involved in transcriptional regulation, RNA processing, and telomere maintenance [5,16,17,18,19,20]. While G4s may play regulatory roles, their accumulation can be deleterious to cells, as they can impede replication, repair, and transcription [16]. Consequently, G4s have emerged as attractive therapeutic targets. The G4-stabilizing compound CX-5461 induces DNA breaks and induces replication fork collapse, highlighting its potential in targeting tumor cells [21].

A key vulnerability of G4-forming regions is their susceptibility to oxidative stress. Guanine has the lowest oxidation potential of the four DNA bases, and its propensity to be oxidized increases when the guanines are stacked or arranged in consecutive runs [22,23]. Oxidation of guanine can produce 8-oxoguanine (8oxoG), a lesion that can mispair with either adenine or cytosine, leading to G to T transversion mutations if left unrepaired [24,25]. Although G4s containing 8oxoG (i.e., 8oxoG4s) can still form, the lesion reduces their stability, rendering them susceptible to nuclease digestion and potentially leading to telomere shortening [26,27]. Nuclear magnetic resonance (NMR) studies have shown that 8oxoG incorporation into human telomeric G4s alters their topology by changing the guanine glycosidic angles from anti to syn configuration [26]. Thus, G4s and 8oxoG4s must be resolved in dividing cells to maintain genomic integrity.

FANCJ (Fanconi anemia complementation group J) is one of several proteins that can unfold G4s in human cells. As an iron–sulfur helicase, FANCJ participates in the repair of interstrand DNA crosslinks as part of the Fanconi anemia pathway [28]. Mutations in FANCJ are associated with increased cancer risk and Fanconi anemia [28,29,30]. FANCJ contributes to genome stability not only through DNA repair but also by resolving G4 structures that impede replication [31,32,33,34,35]. Upon encountering a G4-stalled replication fork, FANCJ binds and unwinds the G4, allowing DNA replication to resume, potentially through the recruitment of translesion polymerases such as REV1 [36] (Figure 2). 

A related RNA helicase RHAU (DHX36) also recognizes G4 structures via an AKKQ motif, where the tandem lysine residues facilitate G4 binding [11,37]. Interestingly, FANCJ contains a similar G4 interaction site with an AKKQ sequence centered around K141 and K142, and alanine substitutions at those positions eliminated G4 binding in the full-length helicase [35]. We showed in our previous work that a FANCJ peptide alone with this motif can target both G4 and 8oxoG4 DNA independently (Figure 3) [36]. In this study, we further characterize the sequence- and damage-dependent recognition of G4 and 8oxoG4 by the FANCJ AKKQ peptide by examining how intervening loop length and guanine oxidation at different positions affect G4-stability, G4 folding, and peptide binding. 

## 2. Results

### 2.1. G-Quadruplex Conformation and Thermal Stability

Circular dichroism (CD) spectroscopy was used to assess the conformation and thermal stability of twelve G4 sequences in buffers containing either KCl or NaCl. As seen in the Appendix A, G4 structures exhibit distinct CD signatures based on their topology, enabling discrimination between parallel, antiparallel, and hybrid configurations (Appendix A) [9,10]. CD experiments were performed as a function of temperature to derive the T_m_ (melting temperature), which reflects the thermal stability of each G4 substrate (Appendix A) [38]. 

Three G4-forming sequences were selected based on their biological relevance and ssDNA loop length, including (GGGT)_4_, a compact parallel G4-forming aptamer for HIV-1 integrase that possesses a single thymine base in the loop [7,39]; (TTAGGG)_4_, the human telomeric repeat sequence that has a TTA loop between the guanine tracts [19]; and (GGGTT)_4_, an artificial sequence with an intermediate loop length (TT) compared to the other two substrates. Oxidative damage was introduced by substituting an 8oxoG at the first (8oxo1), third (8oxo3), or fifth (8oxo5) guanine position of each sequence (Figure 4), thereby targeting each of the three tetrad stacks. The 8oxo1 position disrupts the bottom tetrad, while 8oxo3 and 8oxo5 affect the top and middle tetrads, respectively.

In 150 mM KCl, all (GGGT)_4_ sequences formed a parallel quadruplex (Figure 5A), with only subtle changes to the CD spectra when 8oxoG was incorporated. Due to the temperature limitations of the instrument, the native sequence was not fully unfolded even at 95 °C, and its T_m_ was estimated to be >88.4 ± 8.4 °C (Table 1), consistent with prior reports [7,36]. All 8oxoG substitutions led to reduced thermal stability, with the largest destabilization observed for (GGGT)_4_ 8oxo1 (ΔT_m_ = −19.0 ± 8.4 °C) and the smallest for 8oxo3 (ΔT_m_ = −8.1 ± 8.4 °C).

(GGGTT)_4_ also adopted a parallel conformation in KCl (Figure 5C). While 8oxo1 and 8oxo5 substitutions retained a predominantly parallel fold with minor spectral shifts, the 8oxo3 variant formed an antiparallel quadruplex. The native (GGGTT)_4_ had a T_m_ of 83.5 ± 0.7 °C. 8oxoG destabilized the structure in a similar trend, with 8oxo1 producing the largest drop (ΔT_m_ = −31.5 ± 1.6 °C) and 8oxo3 the smallest (ΔT_m_ = −26.3 ± 0.7 °C).

(TTAGGG)_4_ folded as a hybrid G4 in KCl (Figure 5E). (TTAGGG)_4_ 8oxo1 and (TTAGGG)_4_ 8oxo5 sequences also formed hybrid conformations, although with significant structural changes. (TTAGGG)_4_ 8oxo1 had a significant right shift in the 295 nm peak and a double trough at 260 and 240 nm. (TTAGGG)_4_ 8oxo5 experienced depression in its peak at 295 nm and an increase in its shoulder at 260 nm and trough at 240 nm. Similarly to (GGGTT)_4_ 8oxo3, the (TTAGGG)_4_ 8oxo3 sequence changed to the antiparallel conformation. (TTAGGG)_4_ had a T_m_ of 62.5 ± 0.5 °C (Figure 5F; Table 1). Consistent with a previous study, (TTAGGG)_4_ 8oxo5 had the highest decrease in thermal stability of ΔT_m_ = −23.6 ± 0.8 °C, while (TTAGGG)_4_ 8oxo1 had the lowest change in thermal stability of ΔT_m_ = −8.0 ± 0.7 °C [27].

In NaCl-containing buffer, (GGGT)_4_ maintained a parallel configuration across all damaged variants (Figure 6A). The native sequence had a T_m_ of 70.3 ± 0.7 °C. Substitution with 8oxoG again decreased thermal stability, most notably at the 8oxo1 position (ΔT_m_ = −27.8 ± 0.9 °C), with 8oxo3 being the least affected (ΔT_m_ = −13.9 ± 0.8 °C) (Table 2).

(GGGTT)_4_ retained a parallel conformation in NaCl, but all three 8oxoG4 variants underwent structural transitions: 8oxo1 and 8oxo3 shifted to antiparallel conformations, and 8oxo5 adopted a hybrid-like structure (Figure 6C). Surprisingly, 8oxo1 showed a slight increase in thermal stability (ΔT_m_ = +2.4 ± 1.2 °C), while 8oxo3 had a negligible effect (ΔT_m_ = −0.2 ± 1.2 °C). In contrast, 8oxo5 was the most destabilizing (ΔT_m_ = −17.3 ± 1.4 °C). The 8oxo5 melting curve was acquired from −5 °C to 75 °C due to incomplete folding at 15 °C.

(TTAGGG)_4_ and its 8oxoG4 variants formed antiparallel conformations in NaCl (Figure 6E). The damaged substrates all showed slight peak shifts in the CD spectra compared to the native sequence, indicating slight folding differences. The T_m_ of (TTAGGG)_4_ was determined to be 52.1 ± 0.3 °C (Figure 6F; Table 2). Concurrent with the previous literature and the KCl data above, (TTAGGG)_4_ 8oxo5 had the highest decrease in thermal stability of ΔT_m_ = −18.8 ± 0.5 °C, while (TTAGGG)_4_ 8oxo1 had the lowest of ΔT_m_ = −4.3 ± 0.7 °C [27].

### 2.2. FANCJ AKKQ Binding to 8oxoG4s

To assess the binding of the FANCJ AKKQ motif to G4 and 8oxoG4 structures, fluorescence titrations were performed using a synthetic peptide that corresponds to amino acid residues 129 to 147 of the FANCJ helicase (PEKTTLAAKLSAKKQASIW), except the tyrosine at position 147 was replaced with a tryptophan (Figure 7A). Tryptophan florescence intensity was quenched upon binding to the DNA substrate, and the extent of quenching was plotted against G4 concentration to produce binding isotherms (Figure 7B). The curves were well-described by a 1:1 binding model (Figure 7C). In 150 mM KCl, the FANCJ AKKQ peptide showed preferential binding to oxidatively damaged (GGGT)_4_ sequences (Figure 8; Table 3), with the strongest interaction with 8oxo1 (K = 7.7 ± 0.7 × 10^5^ M^−1^), followed by 8oxo5 (K = 4.4 ± 0.6 × 10^5^ M^−1^), 8oxo3 (K = 3.2 ± 0.1 × 10^5^ M^−1^), and the weakest with the undamaged native sequence (K = 2.7 ± 0.4 × 10^5^ M^−1^). Because the (GGGT)_4_ substrate contained only a single thymine nucleotide loop, we next examined the AKKQ peptide binding to G4 structures formed by (GGGTT)_4_ to test the effects of loop length. All 8oxoG4 variants of this substrate bound to the peptide with similar affinity values (K ≈ 8–9 × 10^5^ M^−1^), which were stronger than its interaction with the undamaged sequence (K = 6.0 ± 0.9 × 10^5^ M^−1^). For the human telomeric G4 (TTAGGG)_4_, this substrate contained an incrementally longer TTA loop; the peptide bound with comparable affinity to all variants of this sequence, including the native structure (K ≈ 1.3–1.4 × 10^6^ M^−1^), suggesting a minimal influence of oxidative damage to binding. 

In NaCl, a similar trend was observed for the (GGGT)_4_ series (Figure 9; Table 4). The AKKQ peptide bound identically and strongest to 8oxo1 (3.4 ± 0.5 × 10^5^ M^−1^) and 8oxo5 (3.4 ± 0.5 × 10^5^ M^−1^), followed by 8oxo3 (2.9 ± 0.5 × 10^5^ M^−1^) and the native sequence (2.3 ± 0.3 × 10^5^ M^−1^). For (GGGTT)_4_, all sequences exhibited similar binding affinities (K ≈ 4.9–5.6 × 10^5^ M^−1^), except for 8oxo1, which showed slightly weaker binding (K = 3.9 ± 1.3 × 10^5^ M^−1^). For the human telomeric (TTAGGG)_4_, the highest affinity was observed for 8oxo5 (K = 1.1 ± 0.1 × 10^6^ M^−1^), followed by the native and 8oxo1 sequences (K = 6.5 × 10^5^ M^−1^), with 8oxo3 binding being the weakest (K = 5.1 ± 1.1 × 10^5^ M^−1^).

## 3. Discussion

### 3.1. G4 Conformation and Stability

Our CD data confirm and expand upon prior studies demonstrating that the human telomeric sequence (TTAGGG)_4_ adopts hybrid and antiparallel conformations in KCl and NaCl, respectively, and that 8oxoG modifications influence both structure and stability in a position-dependent manner [26,27]. As seen by Vorlícková et al., the 8oxo5 substitution produced the greatest thermal destabilization in both KCl and NaCl environments, whereas 8oxo1 was the least disruptive [27].

This positional effect was also evident in the (GGGTT)_4_ series in NaCl, where the native sequence maintained a parallel conformation while the 8oxoG-modified variants adopted antiparallel or hybrid structures. In contrast, the (GGGT)_4_ sequences maintained parallel topology across all variants and salt conditions, with 8oxo1 substitutions consistently producing the greatest destabilization and 8oxo3 the least. A similar pattern was observed for (GGGTT)_4_ in KCl, where only the 8oxo3 variant deviated from the parallel configuration.

Collectively, these findings suggest that the thermal stability of G4 structures in the presence of 8oxoG depends on both the specific guanine position modified and the overall structural context of the G4. In compact, parallel G4s, damage to the first guanine (in the bottom tetrad) was most destabilizing, while damage to the third guanine (top tetrad) had the least effect. Conversely, in more flexible hybrid or antiparallel G4s, the middle tetrad (fifth guanine) was most susceptible to destabilization, while the first guanine remained relatively unaffected. This aligns with earlier NMR-based studies showing significant destabilization and conformational change upon 8oxoG substitution at the fifth guanine in human telomeric G4s [26]. Our data also reinforce the known effects of monovalent cations on G4 stability. Overall, thermal stability was higher in KCl than in NaCl, consistent with the greater stabilizing effect of potassium ions due to their optimal ionic radius for coordinating G-quartets [2,3,4,40,41,42]. Additionally, a trend emerged linking loop length with thermal stability in which shorter loop sequences exhibited higher melting temperatures than longer loops, consistent with previous observations [4,40]. A similar trend was also present but less evident in NaCl buffer, with (GGGT)_4_ and (GGGTT)_4_ having the highest and lowest T_m_ values, respectively. 

### 3.2. FANCJ AKKQ Binding to 8oxoG4s

The FANCJ AKKQ motif is a G4-interaction site, and its derivatives were recently used as tools for visualizing G4s in cells; other peptide variants were also developed to enhance binding to the human telomeric G4 sequence [14,43]. Our fluorescence titration data indicate that the FANCJ AKKQ site can also recognize 8oxoG-modified G4 structures. These interactions were specific and not due to incorporation of the tryptophan fluorophore because experiments in which the AKKQ sequence was replaced with AAAA showed no binding (i.e., fluorescence quenching), even at saturating (TTAGGG)_4_, (GGGT)_4_, and (GGGTT)_4_ concentrations (Appendix A). The binding selectivity was influenced by DNA sequence, structural conformation, and position of the 8oxoG. For the (GGGT)_4_ series, which formed highly stable parallel G4s, AKKQ showed the strongest affinity for 8oxoG-modified sequences—particularly 8oxo1—mirroring the pattern of thermal destabilization. This suggests that the peptide bound preferentially to the damaged base, possibly recognizing features specific to the damaged tetrad rather than to the single nucleotide loop. Since CD spectra for these sequences showed minimal changes in global topology, direct recognition of 8oxoG may drive this binding preference.

In contrast, the AKKQ peptide bound similarly to all 8oxoG variants of the (GGGTT)_4_ sequence, which retained parallel conformations for most of the modifications. The CD spectra revealed structural perturbations and decreased thermal stability of 8oxo1 and 8oxo5 (Figure 5C), but this had no influence on AKKQ binding. We attribute this to the peptide having direct access to the intervening sequencing, supporting a ssDNA loop targeting mechanism rather than direct 8oxoG recognition. 

The telomeric G4 (TTAGGG)_4_ was more conformationally dynamic, and again the AKKQ peptide bound with the same affinity to both damaged and undamaged forms. This is again consistent with the TTA ssDNA loops being more accessible to the peptide in the hybrid and antiparallel configurations, allowing the FANCJ peptide to bind directly to the loop rather than to the 8oxoG. Notably, in KCl, AKKQ binding to (TTAGGG)_4_ was approximately an order of magnitude stronger than for the other sequences, but this distinction diminished in NaCl, except for the 8oxo5 variant, which showed enhanced binding.

One possible explanation for the observed differences in binding preferences is that high sodium concentrations affect the electrostatic interactions between the peptide and the G4s. Sodium ions are known to provide less stabilization to G4s compared to potassium [2,3,4]. We anticipated the FANCJ AKKQ peptide to adopt an alpha helical structure due to previous NMR studies of the structurally and sequentially similar AKKQ motif of the G4-binding protein RHAU [11]. Although our CD studies (Appendix A) did not detect any alpha helical structure in either salt, the addition of trifluoroethanol revealed a helical-like structure in both salt conditions. However, the spectra obtained in each salt were not superimposable. This suggests that sodium ions stabilize the peptide to a lesser extent than potassium ions. 

Interestingly, the affinities of the AKKQ peptide for G4 DNA increased with loop length, further supporting direct interactions with the ssDNA loop. Although FANCJ is a helicase involved in DNA repair, our results suggest that the AKKQ binding site may not function as a direct damage sensor but rather as a loop- or structure-selective binding element. When the loop is too short to be recognized, FANCJ is able to target the 8oxoG damage preferentially (Figure 10). 

### 3.3. Biological Implications and Future Directions

FANCJ is thought to coordinate with the translesion synthesis polymerase REV1 during the replication of G4-rich regions [44]. It is plausible that FANCJ and REV1 perform complementary roles, with FANCJ targeting conformationally dynamic G4s with exposed loops, while REV1 preferentially targets the more compact and stable structures. Exploring the G4-binding preferences and structural requirements of REV1 will be essential to delineate their specialized activities during G4 remodeling. This study examined 8oxoG substitutions at only three guanine positions. Future studies should evaluate a broader set of damage positions, including systematic substitutions across all tetrads, to determine whether the observed positional effects on stability and recognition can be generalized. Additionally, incorporating full-length FANCJ helicase for the binding studies instead of just the AKKQ G4-targeting site would provide insights into whether other FANCJ domains participate in the recognition of 8oxoG4s. 

## 4. Materials and Methods

### 4.1. Buffers and Reagents

All buffers were prepared with reagent-grade chemicals and Type I ultrapure water from a Smart2Pure 6 UV/UF system (ThermoFisher, Waltham, MA, USA). Solutions were sterilized by filtration through 0.22 μm PES membranes prior to use.

### 4.2. Peptide and DNA Oligos 

FANCJ AKKQ peptide (129-PEKTTLAAKLSAKKQASIW-147) was synthesized by Genscript (Piscataway, NJ, USA), and the lyophilized powder was dissolved in 20 mM HEPES with pH 7.5, 150 mM KCl or NaCl, 5 mM TCEP, and 5% (*v*/*v*) glycerol. Peptide concentration was determined using a NanoDrop One UV-Vis microvolume spectrophotometer (ThermoFisher, Waltham, MA, USA), based on the calculated molar extinction coefficients. Peptides were aliquoted, frozen, and stored at −20 °C. 

Oligodeoxyribonucleotides were purchased from Integrated DNA Technologies (IDT, Coralville, IA, USA). DNA concentrations were similarly measured by UV absorbance. All oligonucleotides were stored at 4 °C. A list of the G4 DNA sequences used is provided below in Table 5.

### 4.3. CD Spectroscopy

CD measurements were performed using a JASCO J-815 spectropolarimeter (JASCO, Easton, MD, USA) equipped with a PTC-423S Peltier temperature controller and a Koolance liquid cooling system. G4 samples were dialyzed into 20 mM boric acid containing either 150 mM KCl or NaCl. CD spectra were collected from 220 nm to 320 nm at 25 °C using 0.3 mg/mL DNA. Five traces per sample were averaged, and buffer baselines were subtracted. 

For thermal denaturation experiments, samples were pre-heated to 95 °C and allowed to cool slowly overnight to 25 °C to ensure proper folding. Melting curves were recorded from 15 °C to 95 °C by monitoring the ellipticity at 262 nm (for parallel conformations) or 295 nm (for antiparallel or hybrid structures). Three replicates were performed for each experimental condition, and the resulting data were averaged.

Melting temperatures (***T**_m_***) were determined in SigmaPlot using a two-state transition model described by Greenfield [45], based on the Gibbs–Helmholtz equation: (1)$$∆G=∆H\left(1-\frac{T}{T_{m}}\right)-∆C_{p}(\left(T_{m}-T\right)+T\left(\ln\left(\frac{T}{T_{m}}\right)\right))$$(2)$$K=e^{\left(-\frac{∆G}{RT}\right)}$$(3)$$y=\frac{K}{1+K}$$(4)$${\left[\theta \right]}_{t}=\left(\left({\left[\theta \right]}_{F}-{\left[\theta \right]}_{U}\right)*y\right)+{\left[\theta \right]}_{U}$$
where Δ***G*** represents the change in Gibbs free energy, Δ***H*** is the change in enthalpy, Δ***C_p_*** represents the change in heat capacity, ***R*** is the gas constant, ***K*** represents the equilibrium constant for folding, and ***y*** represents the fraction of folded DNA at any given temperature, ***T***. ***θ_t_*** is the observed ellipticity at any temperature, ***θ_F_*** is the observed ellipticity of the folded DNA, and ***θ_U_*** is the observed ellipticity of the unfolded DNA. The melting temperature, ***T_m_***, of the DNA can then be calculated as the temperature, where ***y*** = 0.5.

### 4.4. Fluorescence Spectroscopy

Fluorescence titration experiments were performed using a Cary Eclipse Fluorescence Spectrophotometer (Agilent Technologies, Santa Clara, CA, USA) equipped with a PCB 1500 Peltier-controlled cuvette holder. Samples were excited at 280 nm, and the emission spectra were collected from 300 to 450 nm at 25 °C. 

A scan of buffer alone, 20 mM HEPES with pH 7.5, 150 mM KCl or NaCl, 5 mM TCEP, and 5% (*v*/*v*) glycerol was used as a baseline. FANCJ AKKQ peptide (5 μM) was titrated with additions of a DNA substrate. The mixture was allowed to equilibrate for 3 min before taking each spectra scan. 

Observed fluorescence quenching (Δ***F***) was calculated using the following equation, with ***F*_0_** representing total fluorescence from the FANCJ peptide alone and ***F_i_*** representing the total fluorescence after the ith addition of DNA:(5)$$∆F=\frac{F_{0}-F_{i}}{F_{0}}$$

Binding isotherms were calculated by plotting Δ***F*** versus total DNA concentration. The resulting curve was fit to a 1:1 interaction model using Scientist 2.0 software (Micromath, St. Louis, MO, USA) to calculate an equilibrium ***K***. The equilibrium ***K*** was calculated using the formulas below, where ***A*** was the amplitude of fluorescence quenching and ***K*** was the equilibrium association constant. ***D_f_*** and ***D_t_*** were the free and total concentration of DNA, while ***P_f_*** and ***P_t_*** described the free and total peptide concentration. Fluorescence titrations were performed in triplicate. The reported ***K*** values were determined from an average of three independent datasets. (6)$$∆F=A\left(\frac{KD_{f}}{1+KD_{f}}\right)$$(7)$$D_{t}=D_{f}(1+KP_{f})$$(8)$$P_{t}=P_{f}(1+KD_{f})$$

## Figures and Tables

**Figure 1 molecules-30-03424-f001:**
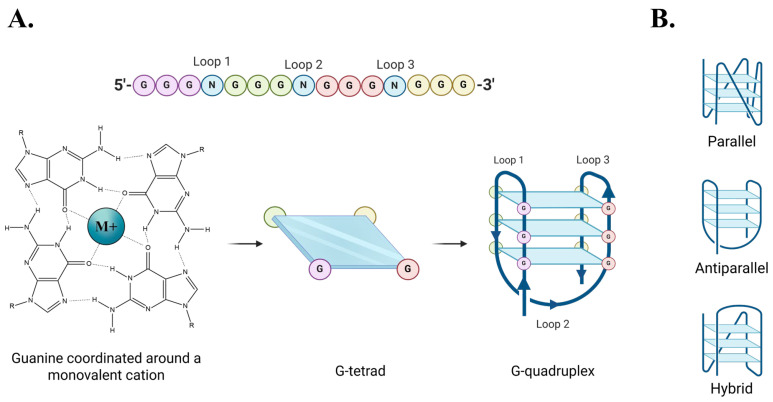
G-quadruplex formation and conformations. (**A**) Formation of a G4 structure. (**B**) Graphic representation of three different intra-strand G4 conformations.

**Figure 2 molecules-30-03424-f002:**
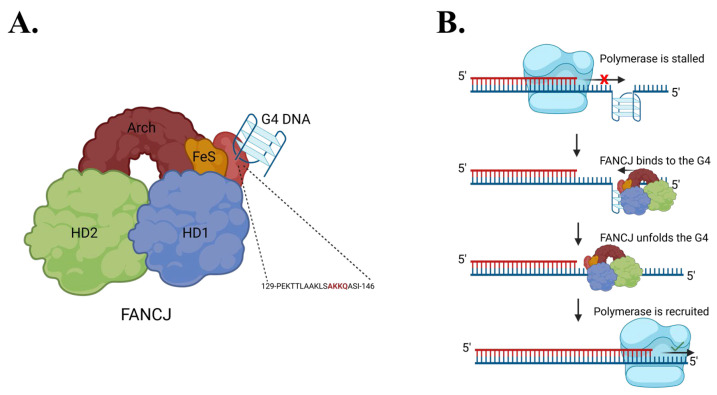
Molecular models of FANCJ. (**A**) FANCJ possesses a G4 recognition site capable of binding G4s. (**B**) A current model for FANCJ-supported DNA replication in G4-rich sequences. DNA polymerase is stalled upon encountering G4 DNA. FANCJ binds to the G4 and unwinds it before a repair polymerase is recruited to continue DNA replication.

**Figure 3 molecules-30-03424-f003:**
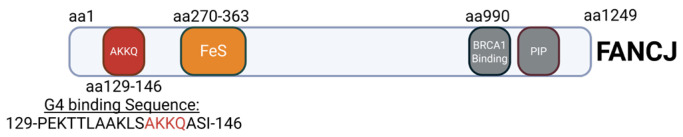
Map of FANCJ. The AKKQ motif within the G4-binding sequence (amino acids 129-146) has been highlighted.

**Figure 4 molecules-30-03424-f004:**
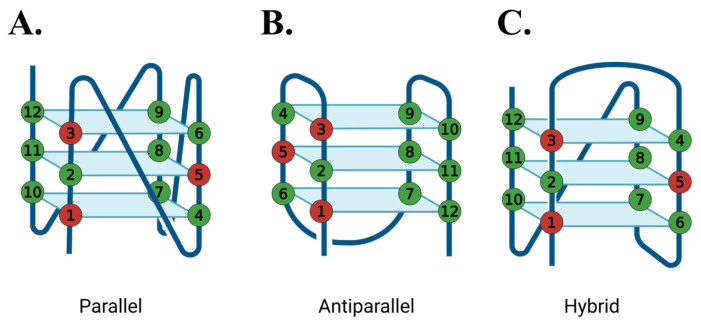
Graphic representation of 8oxoG-modified bases substituted at the first guanine (8oxo1), third guanine (8oxo3), or fifth guanine (8oxo5) of (**A**) parallel, (**B**) antiparallel, and (**C**) hybrid G4 conformations.

**Figure 5 molecules-30-03424-f005:**
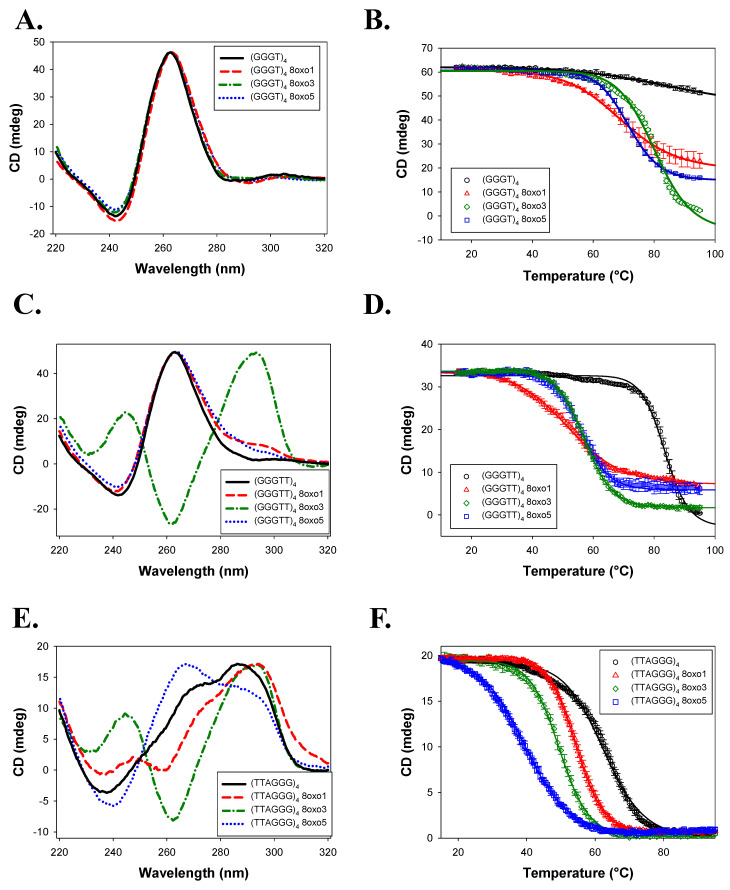
Experiments performed in the presence of 150 mM KCl. (**A**) CD spectra of (GGGT)_4_, (GGGT)_4_ 8oxo1, (GGGT)_4_ 8oxo3, and (GGGT)_4_ 8oxo5 taken from 320 to 220 nm. (**B**) Temperature melt of same sequences seen in 5A. Melt taken at 262 nm from 15 to 95 °C. (**C**) CD spectra of (GGGTT)_4_, (GGGTT)_4_ 8oxo1, (GGGTT)_4_ 8oxo3, and (GGGTT)_4_ 8oxo5 taken from 320 to 220 nm. (**D**) Temperature melt of same sequences seen in 5C. Melt taken at 262 nm for all sequences except (GGGTT)_4_ 8oxo3, which was taken at 295 nm. (**E**) CD spectra of (TTAGGG)_4_, (TTAGGG)_4_ 8oxo1, (TTAGGG)_4_ 8oxo3, and (TTAGGG)_4_ 8oxo5 taken from 320 to 220 nm. (**F**) Temperature melt of the same sequences seen in 5E. Melt taken at 295 nm.

**Figure 6 molecules-30-03424-f006:**
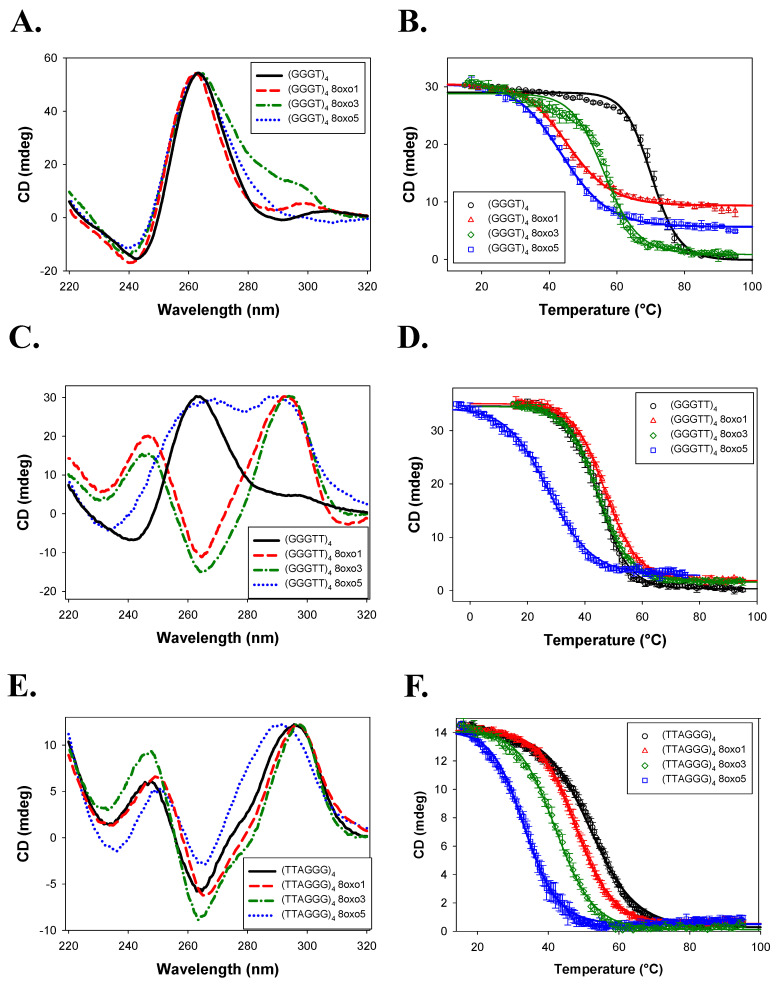
Experiments performed in the presence of 150 mM NaCl. (**A**) CD spectra of (GGGT)_4_, (GGGT)_4_ 8oxo1, (GGGT)_4_ 8oxo3, and (GGGT)_4_ 8oxo5 taken from 320 to 220 nm. (**B**) Temperature melt of same sequences as seen in 6A. Melt taken at 262 nm from 15 to 95 °C. (**C**) CD spectra of (GGGTT)_4_, (GGGTT)_4_ 8oxo1, (GGGTT)_4_ 8oxo3, and (GGGTT)_4_ 8oxo5 taken from 320 to 220 nm. (**D**) Temperature melt of same sequences as seen in 6C. Melt taken at 262 nm for all sequences except (GGGTT)_4_ 8oxo3, which was taken at 295 nm. (**E**) CD spectra of (TTAGGG)_4_, (TTAGGG)_4_ 8oxo1, (TTAGGG)_4_ 8oxo3, and (TTAGGG)_4_ 8oxo5 taken from 320 to 220 nm. (**F**) Temperature melt of the same sequences as seen in 6E. Melt taken at 295 nm.

**Figure 7 molecules-30-03424-f007:**
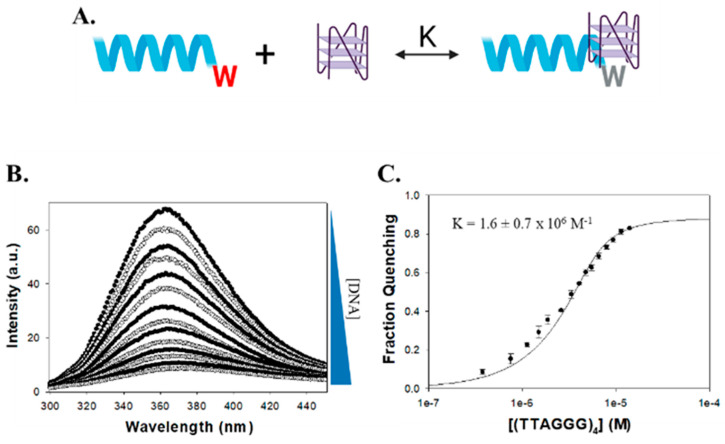
Overview of fluorescence spectroscopy analyses. (**A**) Molecular model of fluorescence spectroscopy titrations. AKKQ is equipped with a tryptophan residue (W) that fluoresces. When bound to DNA, the residue is quenched and fluorescence intensity decreases. (**B**) Titration of AKKQ with (GGGT)_4_. As DNA concentration increases, the peak intensity decreases. (**C**) Analysis of fluorescence titration. The decrease in integrated peak is plotted as a function of DNA concentration. The resulting isotherm is fit to a 1:1 binding interaction model to generate an equilibrium K value.

**Figure 8 molecules-30-03424-f008:**
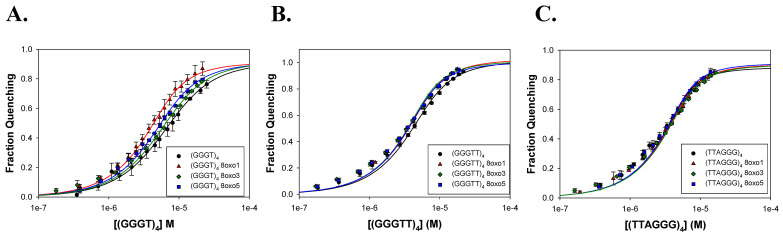
Summary of FANCJ AKKQ-G4 binding in KCl by fluorescence spectroscopy. Trials performed in 20 mM HEPES pH 7.5, 150 mM KCl, 5 mM TCEP, and 5% (*v*/*v*) glycerol. (**A**) FANCJ AKKQ equilibrium binding to (GGGT)_4_ and its respective 8oxoG4 sequences. (**B**) FANCJ AKKQ binding to (GGGTT)_4_ and its respective 8oxoG4 sequences. (**C**) FANCJ AKKQ binding to (TTAGGG)_4_ and its respective 8oxoG4 sequences.

**Figure 9 molecules-30-03424-f009:**
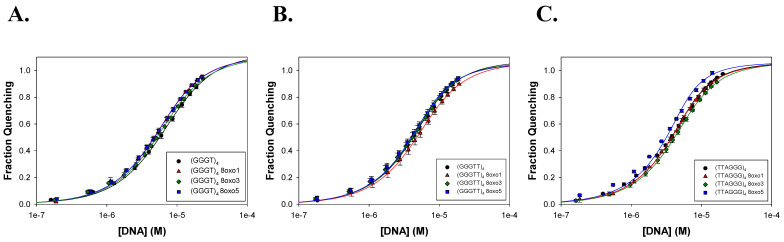
Summary of FANCJ AKKQ-G4 binding in NaCl by fluorescence spectroscopy. Trials performed in 20 mM HEPES pH 7.5, 150 mM NaCl, 5 mM TCEP, and 5% (*v*/*v*) glycerol. (**A**) FANCJ AKKQ binding to (GGGT)_4_ and respective 8oxoG4 sequences. (**B**) FANCJ AKKQ binding to (GGGTT)4 and respective 8oxoG4 sequences. (**C**) FANCJ AKKQ binding to (TTAGGG)4 and respective 8oxoG4 sequences.

**Figure 10 molecules-30-03424-f010:**
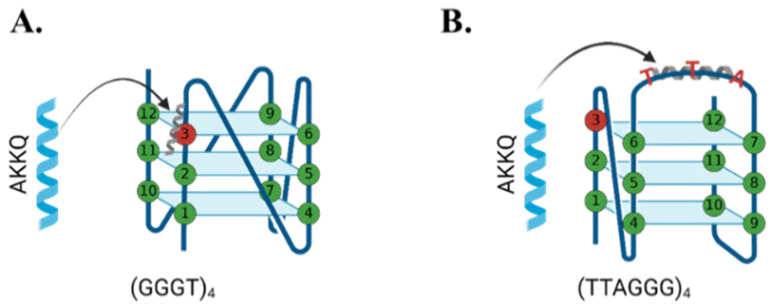
Proposed binding interaction model between FANCJ AKKQ and 8oxoG4 sequences. (**A**) For G4 sequences that form compact structures with short intervening sequences, such as (GGGT)_4_, AKKQ preferentially targets the 8oxoG site and promotes repair. (**B**) For G4 sequences that expose the intervening sequence, as with antiparallel or hybrid conformations, AKKQ preferentially targets the intervening sequence and ignores the 8oxoG.

**Table 1 molecules-30-03424-t001:** Summary of G4 conformation and melting temperature in 150 mM KCl.

Name	Conformation	T_m_ (°C)	ΔT_m_ (°C)
(GGGT)_4_	Parallel	>88.4 ± 8.4	-
(GGGT)_4_ 8oxo1	Parallel	69.4 ± 0.6	−19.0 ± 8.4
(GGGT)_4_ 8oxo3	Parallel	80.3 ± 0.4	−8.1 ± 8.4
(GGGT)_4_ 8oxo5	Parallel	71.6 ± 0.4	−16.8 ± 8.4
(GGGTT)_4_	Parallel	83.5 ± 0.7	-
(GGGTT)_4_ 8oxo1	Parallel	52.0 ± 1.4	−31.5 ± 1.6
(GGGTT)_4_ 8oxo3	Antiparallel	57.2 ± 0.3	−26.3 ± 0.7
(GGGTT)_4_ 8oxo5	Parallel	55.9 ± 0.3	−27.6 ± 0.8
(TTAGGG)_4_	Hybrid	62.5 ± 0.5	-
(TTAGGG)_4_ 8oxo1	Hybrid-esque	54.5 ± 0.5	−8.0 ± 0.7
(TTAGGG)_4_ 8oxo3	Antiparallel	49.5 ± 0.3	−13.0 ± 0.6
(TTAGGG)_4_ 8oxo5	Hybrid-esque	38.9 ± 0.6	−23.6 ± 0.8

**Table 2 molecules-30-03424-t002:** Summary of G4 conformation and melting temperature in 150 mM NaCl.

Name	Conformation	T_m_ (°C)	ΔT_m_ (°C)
(GGGT)_4_	Parallel	70.3 ± 0.7	-
(GGGT)_4_ 8oxo1	Parallel	42.5 ± 0.5	−27.8 ± 0.9
(GGGT)_4_ 8oxo3	Parallel	56.4 ± 0.4	−13.9 ± 0.8
(GGGT)_4_ 8oxo5	Parallel	44.0 ± 0.3	−26.3 ± 0.8
(GGGTT)_4_	Parallel	44.8 ± 1.2	-
(GGGTT)_4_ 8oxo1	Antiparallel	47.2 ± 0.3	2.4 ± 1.2
(GGGTT)_4_ 8oxo3	Antiparallel	44.6 ± 0.2	−0.2 ± 1.2
(GGGTT)_4_ 8oxo5	Hybrid-esque	27.5 ± 0.9	−17.3 ± 1.4
(TTAGGG)_4_	Antiparallel	52.1 ± 0.3	-
(TTAGGG)_4_ 8oxo1	Antiparallel	47.8 ± 0.6	−4.3 ± 0.7
(TTAGGG)_4_ 8oxo3	Antiparallel	42.2 ± 0.7	−9.9 ± 0.8
(TTAGGG)_4_ 8oxo5	Antiparallel	33.3 ± 0.4	−18.8 ± 0.5

**Table 3 molecules-30-03424-t003:** Summary of FANCJ AKKQ-G4 binding by fluorescence spectroscopy in 150 mM KCl.

Name	K (M^−1^)	R^2^
GGGT)_4_	2.7 ± 0.4 × 10^5^	0.997
(GGGT)_4_ 8oxo1	7.7 ± 0.7 × 10^5^	0.998
(GGGT)_4_ 8oxo3	3.2 ± 0.1 × 10^5^	0.997
(GGGT)_4_ 8oxo5	4.4 ± 0.6 × 10^5^	0.997
(GGGTT)_4_	6.0 ± 0.9 × 10^5^	0.997
(GGGTT)_4_ 8oxo1	8.2 ± 1.0 × 10^5^	0.997
(GGGTT)_4_ 8oxo3	9.6 ± 1.0 × 10^5^	0.996
(GGGTT)_4_ 8oxo5	9.3 ± 0.8 × 10^5^	0.997
(TTAGGG)_4_	1.6 ± 0.7 × 10^6^	0.996
(TTAGGG)_4_ 8oxo1	1.3 ± 0.4 × 10^6^	0.997
(TTAGGG)_4_ 8oxo3	1.2 ± 0.1 × 10^6^	0.996
(TTAGGG)_4_ 8oxo5	1.4 ± 0.5 × 10^6^	0.997

**Table 4 molecules-30-03424-t004:** Summary of FANCJ AKKQ-G4 binding by fluorescence spectroscopy in 150 mM NaCl.

Name	K (M^−1^)	R^2^
(GGGT)_4_	2.3 ± 0.3 × 10^5^	0.9997
(GGGT)_4_ 8oxo1	3.4 ± 0.5 × 10^5^	0.9996
(GGGT)_4_ 8oxo3	2.9 ± 0.5 × 10^5^	0.9994
(GGGT)_4_ 8oxo5	3.4 ± 0.5 × 10^5^	0.9997
(GGGTT)_4_	4.9 ± 2.0 × 10^5^	0.9995
(GGGTT)_4_ 8oxo1	3.9 ± 1.3 × 10^5^	0.9993
(GGGTT)_4_ 8oxo3	5.2 ± 1.7 × 10^5^	0.9994
(GGGTT)_4_ 8oxo5	5.6 ± 1.1 × 10^5^	0.9992
(TTAGGG)_4_	6.5 ± 0.6 × 10^5^	0.9991
(TTAGGG)_4_ 8oxo1	6.5 ± 1.4 × 10^5^	0.9996
(TTAGGG)_4_ 8oxo3	5.1 ± 1.1 × 10^5^	0.9998
(TTAGGG)_4_ 8oxo5	1.1 ± 0.1 × 10^6^	0.9974

**Table 5 molecules-30-03424-t005:** DNA sequences.

Name	Sequence (5′→3′)	Molar Extinction Coefficient (M^−1^·cm^−1^)	Molecular Weight (g/mol)
(GGGT)_4_	GGGTGGGTGGGTGGGT	157,200	5105.3
(GGGT)_4_ 8oxo1	T/i8oxodG/GGTGGGTGGGTGGGT	159,100	5425.5
(GGGT)_4_ 8oxo3	GG/i8oxodG/TGGGTGGGTGGGT	151,600	5121.3
(GGGT)_4_ 8oxo5	GGGTG/i8oxodG/GTGGGTGGGT	151,600	5121.3
(GGGTT)_4_	GGGTTGGGTTGGGTTGGGTT	189,600	6322.1
(GGGTT)_4_ 8oxo1	/i8oxodG/GGTTGGGTTGGGTTGGGTT	184,000	6338.1
(GGGTT)_4_ 8oxo3	GG/i8oxodG/TTGGGTGGGTTGGGTT	184,000	6338.1
(GGGTT)_4_ 8oxo5	GGGTTG/i8oxodG/GTTGGGTTGGGTT	184,000	6338.1
(TTAGGG)_4_	TTAGGGTTAGGGTTAGGGTTAGGG	244,600	7575.0
(TTAGGG)_4_ 8oxo1	TTA/i8oxodG/GGTTAGGGTTAGGGTTAGGG	239,000	7590.9
(TTAGGG)_4_ 8oxo3	TTAGG/i8oxodG/TTAGGGTTAGGGTTAGGG	239,000	7590.9
(TTAGGG)_4_ 8oxo5	TTAGGGTTAG/i8oxodG/GTTAGGGTTAGGG	239,000	7590.9

## Data Availability

All raw and processed data files are freely available through Open Science Framework at the link https://osf.io/z4a8q/.

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
