# Peer review of "8-Oxoguanine Disrupts G-Quadruplex DNA Stability and Modulates FANCJ AKKQ Peptide Binding"

_molecules, 2025, doi:10.3390/molecules30163424_

Round 1
Reviewer 1 Report
Comments and Suggestions for Authors
The paper explores the destabilizing effect of 8-OxoG bases within DNA quadruplexes of predefined sequences and examines how this destabilization influences their interaction with the FANCJ AKKQ. While the topic is not particularly novel, the investigation of how substituting guanosines with 8-OxoG affects target interaction could be of significant interest.
In my opinion, this is the central focus of the study. However, I believe that the brief description of the interactions between the modified GQ structures and the AKKQ peptide is insufficient for publication in Molecules.
The authors should consider the following revisions before resubmitting the manuscript:
- Figure 4 is unnecessary, especially panels B and C, which may be misleading. These could be included in the Supplementary Information (SI).
- Figure 5, which depicts the model used to represent the position of 8-OxoG, is misleading because it does not accurately reflect the structures studied. It always shows a parallel GQ, whereas the table reports a hybrid or antiparallel structure—for example, ((GGGTT)4 8oxo3) and a hybrid structure for ((TTAGGG)4 8oxo1), depending on the sequence (see Table 1).
- In evaluating the relationship between the position of 8-OxoG within the sequences and the binding affinity to the model peptide, a peptide with a tryptophan at the end was synthesized. However, studies demonstrating the absence of the terminal tryptophan’s role are not included. A control experiment with a different synthetic peptide containing tryptophan should be performed to exclude possible interactions.
- Given the paper’s focus on the interaction between modified GQs and the peptide, more detailed structural data should be provided. For example, NMR titrations could be performed, which might also eliminate the need for tryptophan in the peptide.
- The bibliography contains too few references from the past 10 yearsAn updated and more comprehensive bibliography would make the paper more engaging for Molecules readers.
Author Response
- “Figure 4 is unnecessary, especially panels B and C, which may be misleading.” We have moved Figure 4 to SI since the CD profiles of parallel, antiparallel, and hybrid G4 structures are already well-established. We agree with the reviewer that the Figure could be removed entirely but we kept it has a supplementary figure for easy reference.
-
“Figure 5, which depicts the model used to represent the position of 8-OxoG, is misleading because it does not accurately reflect the structures studied.” This is an excellent suggestion. We have updated to figure to show where the 8oxo1, 8oxo3, and 8oxo5 positions are in parallel, antiparallel, and hybrid G4 structures rather than using the same parallel G4 in the cartoon.
-
“In evaluating the relationship between the position of 8-OxoG within the sequences and the binding affinity to the model peptide, a peptide with a tryptophan at the end was synthesized. However, studies demonstrating the absence of the terminal tryptophan’s role are not included. A control experiment with a different synthetic peptide containing tryptophan should be performed to exclude possible interactions.” Normally a tyrosine residue occupies this position so a tryptophan substitution is a modest change and it was included to serve as a fluorophore for detecting DNA binding. We agree with the reviewer’s comment, and rather than using a different synthetic peptide, we performed a negative control experiment in which the AKKQ sequence was modified to AAAA (still retaining the tryptophan) and showed that it did not bind to the three wt G4 sequences. We did not use the 8oxoG substrates for full titrations because they were cost-prohibitive, but this data showed that the G4-binding activity originated from the AKKQ motif and not the tryptophan. This panel is now included in Fig S2.
-
“Given the paper’s focus on the interaction between modified GQs and the peptide, more detailed structural data should be provided. For example, NMR titrations could be performed, which might also eliminate the need for tryptophan in the peptide.” This is an excellent suggestion but our lab does not have the capability or resources to pursue NMR experiments at this time. We had explored using ITC to examine the peptide-G4 interactions so that the tryptophan was not needed, and the native tyrosine residue still enabled us to reliably measure the peptide concentration by UV-Vis. However, the ITC experiments had poor signal:noise and we were able to obtain much more rigorous binding data by fluorescence spectroscopy.
- “The bibliography contains too few references from the past 10 years. An updated and more comprehensive bibliography would make the paper more engaging for Molecules readers.” We have included more recent citations. CD analysis of G4s is well-established but the discovery of the AKKQ binding motif is still relatively novel and there has been little research done with this peptide. However, we included more recent work by Zheng et al (2020) in which they developed the G4P with the RHAU/DHX36 AKKQ peptide to visualize G4s in cells, as well as Gaur et al (2023) in which they tested different FANCJ AKKQ peptide variants to improve the binding to the human telomeric G4 DNA.
Reviewer 2 Report
Comments and Suggestions for Authors
Laura Campbell and co-authors reported here 8oxoG play a key role in regulating the unsolve function of FANCJ to G-quadruplex DNA molecules. It’s interesting to find that the structure and stability of G-quadruplxe DNA was changed in the presence of 8oxoG in varied G-quadruplex DNA sequence, as well as the binding affinity of FANCJ AKKQ to G-quadruplex DNA. However, there are still some issues should be addressed.
- Considering that FANCJ plays a key role in resolving G-quadruplex DNA to allow DNA replication to resume, the authors should provid convinced evidence to show that 8oxoGwas occured in the process.
- The authors should provides convinced evidences like FRET to show that FANCJ play a key role in unfold and re-fold of G-quadruplex DNA modified with 8oxoG.
- 2 A is the same as that in Nucleic Acids Research, 8742–8753, 2016(Figure 7A).
Author Response
- “Considering that FANCJ plays a key role in resolving G-quadruplex DNA to allow DNA replication to resume, the authors should provide convinced evidence to show that 8oxoG was occurred in the process.” The DNA substrates we used were synthesized and validated by IDT with 8oxoG at the specified sites. We will make the QC data available in Open Science Framework, but we did not include them in the supplementary information because of the large number of DNA substrates used.
-
“The authors should provide convinced evidences like FRET to show that FANCJ play a key role in unfold and re-fold of G-quadruplex DNA modified with 8oxoG.” This is an excellent suggestion and we are currently using fluorescence-based unfolding assays to study the how full-length FANCJ helicase and REV1 polymerase remodel different 8oxoG4s. For this manuscript, we are solely focusing on the direct binding between the FANCJ AKKQ G4 recognition peptide and the different 8oxoG4 substrates and so there is no unfolding taking place.
- “2A is the same as that in Nucleic Acids Research, 8742–8753, 2016(Figure 7A).” Thank you for pointing this out and we have prepared an original figure to replace the cartoon in Figure 2A.
Round 2
Reviewer 1 Report
Comments and Suggestions for Authors
The authors have made the requested changes satisfactorily.
Author Response
“Considering that FANCJ plays a key role in resolving G-quadruplex DNA to allow DNA replication to resume, the authors should provide convinced evidence to show that 8oxoG was occurred in the process.” The Editor clarified this comment in which the reviewer wanted us to confirm/discuss whether FANCJ is involved in the cellular processes involving 8oxoG. Typically, 8oxoG is repaired by the base excision repair (BER) pathway in which a DNA glycosylase removes the damaged base before a new base is incorporated. FANCJ is not involved in this process unless its helicase activity is needed. The Brosh Lab showed that FANCJ can efficiently unwind 8oxoG-modified DNA duplexes (2009 JBC), which suggested that FANCJ can “open up” DNA structures to expose the 8oxoG damage. In my previous work with the Spies Lab (2016 NAR), we showed that the AKKQ motif in FANCJ was a G4-interaction site, but we did not know at the time whether it can also recognize 8oxoG4s. In this manuscript, we further show that the FANCJ AKKQ motif can directly bind to 8oxoG4s and that the affinities of these interactions depend on the intervening loop length. Taken together, our results suggest that FANCJ can bind and unfold 8oxoG4s. Once the 8oxoG is exposed, it can be removed by the conventional BER pathway by the OGG1 glycosylase. Alternatively, the damage could be bypassed by direct recruitment of a repair polymerase, and we have shown that FANCJ can bind to REV1 (2019 Genes) which may facilitate this activity.
“The authors should provide convinced evidences like FRET to show that FANCJ play a key role in unfold and re-fold of G-quadruplex DNA modified with 8oxoG.” The Editor clarified this comment in which the Reviewer asked whether the G4 is unfolded/folded upon binding to the FANCJ peptide. We have monitored this by CD, and even at saturating concentration of the FANCJ peptide, the G4s and 8oxoG4s remain folded at 25C. As stated in our earlier response, this is an excellent suggestion from the reviewer and we are currently examining the 8oxoG4 unfolding activities of the full-length FANCJ helicase and the catalytic core of the REV1 polymerase. Some helicases/polymerases can disrupt DNA secondary structure upon binding. In fact, our preliminary data shows that FANCJ, even in the absence of ATP can unfold the G4 telomeric sequence, presumably to load the DNA substrate onto its motor domains because a G4 substrate with a T15 ssDNA overhang did not show this activity. In this manuscript, we are only working with the FANCJ AKKQ peptide, without the helicase domains, so it was not surprisingly that the peptide alone could bind to the 8oxoG4s but could not disrupt them.